# Endurance-trained subjects and sedentary controls increase ventricular contractility and efficiency during exercise: Feasibility of hemodynamics assessed by non-invasive pressure-volume loops

Björn Östenson[1], Ellen Ostenfeld[1‡], Jonathan Edlund[1‡], Einar Heiberg[1,2‡], Håkan Arheden[1‡], Katarina Steding-Ehrenborg[1]*

1 Clinical Physiology, Department of Clinical Sciences Lund, Lund University, Skåne University Hospital, Lund, Sweden, 2 Wallenberg Centre for Molecular Medicine, Lund University, Lund, Sweden

☯ These authors contributed equally to this work.
‡ ES, JE, EH and HA also contributed equally to this work.
* katarina.ehrenborg@med.lu.se

**Citation:** Östenson B, Ostenfeld E, Edlund J, Heiberg E, Arheden H, Steding-Ehrenborg K (2023) Endurance-trained subjects and sedentary controls increase ventricular contractility and efficiency during exercise: Feasibility of hemodynamics assessed by non-invasive pressure-volume loops. PLoS ONE 18(5): e0285592. https://doi.org/10.1371/journal.pone.0285592

**Data Availability Statement:** Full data cannot be shared publicly because of the small sample size

## Abstract

### Introduction

Pressure-volume (PV) loops can be used to assess both load-dependent and load-independent measures of cardiac hemodynamics. However, analysis of PV loops during exercise is challenging as it requires invasive measures. Using a novel method, it has been shown that left ventricular (LV) PV loops at rest can be obtained non-invasively from cardiac magnetic resonance imaging (CMR) and brachial pressures. Therefore, the aim of this study was to assess if LV PV loops can be obtained non-invasively from CMR during exercise to assess cardiac hemodynamics.

### Methods

Thirteen endurance trained (ET; median 48 years [IQR 34–60]) and ten age and sex matched sedentary controls (SC; 43 years [27–57]) were included. CMR images were acquired at rest and during moderate intensity supine exercise defined as 60% of expected maximal heart rate. Brachial pressures were obtained in conjunction with image acquisition.

### Results

Contractility measured as maximal ventricular elastance (Emax) increased in both groups during exercise (ET: 1.0 mmHg/ml [0.9–1.1] to 1.1 mmHg/ml [0.9–1.2], p<0.01; SC: 1.1 mmHg/ml [0.9–1.2] to 1.2 mmHg/ml [1.0–1.3], p<0.01). Ventricular efficiency (VE) increased in ET from 70% [66–73] at rest to 78% [75–80] (p<0.01) during exercise and in SC from 68% [63–72] to 75% [73–78] (p<0.01). Arterial elastance ($E_A$) decreased in both groups (ET: 0.8 mmHg/ml [0.7–0.9] to 0.7 mmHg/ml [0.7–0.9], p<0.05; SC: 1.0 mmHg/ml [0.9–1.2] to 0.9 mmHg/ml [0.8–1.0], p<0.05). Ventricular-arterial coupling ($E_A/E_{max}$) also decreased in both

and the risk of identifying individual subjects. Data are available from the Cardiac MR Group in Lund, Lund University, Sweden for researchers who meet the criteria for access to confidential data and after additional consent from the research participants. For data access requests, please contact the Cardiac MR Group in Lund via email: cmr-lund@med.lu.se.

**Funding:** KSE: Swedish Research Council for Sport Science P2021-0052, Swedish Olympic Committee (grant number N/A, Southern Health Care Region (ALF) 2018-Project0123 EH: VR 2017-04389 The funders had no role in study design, data collection and analysis, decision to publish, or preparation of the manuscript.

**Competing interests:** EH is the founder of the company Medviso AB which develops medical image analysis software Segment. KSE has received minor speaker compensation from Bayer Medical. All other authors declare no competing interests.

groups (ET: 0.9 [0.8–1.0] to 0.7 [0.6–0.8], $p<0.01$; SC: 1.0 [0.9–1.1] to 0.7 [0.7–0.8], $p<0.01$).

## Conclusions

This study demonstrates for the first time that LV PV loops can be generated non-invasively during exercise using CMR. ET and SC increase ventricular efficiency and contractility and decrease afterload and ventricular-arterial coupling during moderate supine exercise. These results confirm known physiology. Therefore, this novel method is applicable to be used during exercise in different cardiac disease states, which has not been possible non-invasively before.

## Introduction

Invasive pressure-volume (PV) loops are gold standard to assess hemodynamic function and cardiac performance [1] and can provide information on both load-dependent and load-independent parameters such as myocardial contractility, ventricular efficiency, and arterial elastance [2]. Thus, PV loops may provide additional information in patients with decreased cardiac function and potentially explain the enhanced cardiac function seen in endurance-trained (ET) athletes during exercise [3–8]. However, measuring PV loops during exercise has been challenging, likely due to the invasive nature previously needed to acquire the measurements. A newly developed and experimentally validated non-invasive method to calculate left ventricular (LV) PV loops at rest from cardiac magnetic resonance (CMR) images and brachial pressure has recently been presented [9, 10]. This method has later been validated against invasive measurements in humans [11] and could potentially be used to obtain PV loops also in the setting of exercise. Therefore, the aim of this study was to assess if LV PV loops can be obtained from CMR and thereby assess cardiac hemodynamics non-invasively during exercise.

## Materials and methods

This single-center, exploratory, observational, prospective study was approved by the regional ethics committee in Lund (Dnr 741/2004, with complementary Dnr 269/2005) and follow the Declaration of Helsinki. All participants provided their written consent to participate in the study. The study complies with the STROBE guidelines for cross-sectional observational studies [12].

### Study population

Thirteen ET subjects from a local triathlon association with median age 48 years [interquartile range (IQR) 34–60, one female] and ten healthy sedentary controls (SC) matched for age and sex (age 43 years [27–57], one female) were included. ET were defined as individuals who practice systematic exercise with the intent to improve performance in competition [13]. ET had a weekly training volume of 6 hours [4.4–7.5] over the last six months and had been active in regular endurance training for 15 years [10–20]. Weekly training volume of SC was equal to or less than 150 minutes of aerobic exercise at moderate intensity according to the recommendations for physical activity from World Health Organization 2020 guidelines on physical activity and sedentary behavior [14]. Exclusion criteria were any pre-existing cardiovascular diseases, health problems requiring regular medication, smoking, and contraindications to CMR (claustrophobia, or metal implants and devices). Study participants were recruited during 2020

using public advertisements. The rationale of recruiting ET for this feasibility study was that this study population was expected to be comfortable with multiple cycling bouts for a prolonged duration and breath-holds in an exercised state, minimizing the risk of the subject not being able to complete the exercise protocol. The SC were then recruited to study the impact of endurance training on cardiovascular hemodynamics using the proposed method. Sample size was limited in this pilot analysis to demonstrate feasibility of non-invasive PV loops using CMR.

## Cardiac magnetic resonance imaging and exercise protocol

Cardiac magnetic resonance images were acquired using a 1.5 T scanner (Siemens Aera, Siemens Healthineers, Erlangen, Germany). Cine balanced steady state free precession images were acquired in short-axis view covering the ventricles, as well as in 2-chamber, 3-chamber, and 4-chamber long-axis views with retrospective ECG gating. Cine images were acquired at rest and during cycling exercise with an MR-compatible pedal ergometer (Lode BV, Groningen, The Netherlands) in the supine position. Breath-hold duration was typically 8–12 seconds for each short-axis image acquisition (1–3 slices) and 8–10 seconds for each long-axis image. Typical image parameters were as follows: spatial resolution 1.0x1.0 mm, slice thickness 8 mm, temporal resolution 46 ms at rest and 31 ms during exercise, flip angle 68˚, echo time 1.1 ms, repetition time 41 ms.

Exercise load on the ergometer was manually adjusted using dedicated software (Lode BV, Groningen, The Netherlands). Study participants bicycled during incremental load from a low starting load until achieving moderate exercise intensity, defined as 60% of estimated maximum heart rate calculated as 220 beats per minute minus age [15]. The exercise intensity was chosen to induce an exercise-related effect on hemodynamics, while still minimizing movement artifacts. In the case of unreadable exercise heart rate due to motion disturbances, study participants bicycled until subjective moderate intensity was reached, defined as 14–15 on the Borg rating of perceived exertion scale. Breath-hold cine images were acquired immediately after exercise cessation and, to ensure high heart rates during image acquisition, exercise was interleaved with exercise cessation. Each breath-hold image acquisition was set to last 5–15 seconds by modifying the number of concatenations of the sequence. Shorter breath-holds resulted in fewer acquired short-axis slices per exercise bout compared to longer breath-holds. The number of concatenations was individually modified to adjust for individual breath-hold capacity of the study participants. Typically, one to three short-axis slices were acquired during cessation in the scanner before the next exercise session started. A new exercise bout during incremental load was initiated immediately after image acquisition from a low starting load until target intensity was achieved again. This process was repeated until images of the whole ventricle were acquired. Heart rate was noted for each image slice (S1 Table). Mean exercise heart rate was calculated by summation of heart rates for each imaging slice divided by the number of image slices. For all participants, blood pressure measurement was initiated immediately after exercise cessation using an automatic brachial sphygmomanometer (Invivo Expression, Invivo Corporation, Orlando, FL, USA), followed by breath-hold and image acquisition during the blood pressure measurement. Mean exercise blood pressure was calculated by summation of pressures acquired for each imaging slice acquisition divided by number of acquisitions. Blood pressure values for each exercise bout can be found in S2 Table.

## Data analysis

Image analysis was performed using the software Segment v3.2 R8688 (Medviso AB, Lund, Sweden, http://segment.heiberg.se) [16]. LV endocardial and epicardial borders of all slices in

all time frames during one cardiac cycle were automatically delineated using an AI-based LV segmentation tool [17] with manual corrections by one observer with seven years of CMR experience (BÖ) to obtain time resolved volume curves. Left ventricular mass (LVM) was measured as an internal validation to ensure less than 2 g differed over one cardiac cycle.

## Pressure-volume loop parameters

LV PV loops at rest and exercise were computed using a previously developed and validated plugin analysis tool in the software Segment [10]. In brief, the LV time-varying elastance curve was modeled by the analytical double-Hill equation. The parameters in the double-Hill equation were optimized by data from 875 invasively measured elastance curves. The amplitude of the modeled LV elastance curve is defined by left ventricular end-systolic pressure (LVESP) and end-diastolic pressure (LVEDP). LVESP can be approximated by non-invasive brachial sphygmomanometer measures of systolic blood pressure (SBP) and diastolic blood pressure (DBP) [18]:

$$\text{LVESP} = \frac{2}{3}\text{SBP} + \frac{1}{3}\text{DBP} \tag{1}$$

The plugin requires user-input assumption of LVEDP, which was set to 0–15 mmHg according to data from previous studies [19–21]. LV blood pressure over the cardiac cycle is calculated from the modeled LV elastance curve and time-resolved CMR volumetric measurements, and the final output is the PV loop.

Stroke work (SW), potential energy (PE), pressure-volume area (PVA), ventricular efficiency (VE), energy per ejected volume (EEV), mean external power (MEP), maximal ventricular elastance ($E_{max}$), and effective arterial elastance ($E_A$) were computed as follows:

SW is the mechanical energy transferred from the myocardium to the blood to generate left ventricular stroke volume (LVSV). It was derived from the PV loop as the surface integral of the PV loop.

PE is the mechanical energy stored in the myocardium and dissipating as heat during contraction:

$$\text{PE} = \text{V}\left(\text{T}_{\text{LVP,max}}\right) \times \frac{\text{LVESV} - \text{V}_0}{2} \tag{2}$$

where $\text{V}(_{\text{TLVP,max}})$ is the LV volume at the time-point where the LV pressure is maximal, LVESV is left ventricular end-systolic volume, and $\text{V}_0$ is the volume axis intercept (i.e. the theoretical LV volume when LV pressure is 0 mmHg). $\text{V}_0$ was assumed to be zero mmHg as previously suggested [22].

PVA is the total ventricular mechanical energy of one given heartbeat and is linearly correlated with myocardial oxygen consumption [23, 24]. It was calculated as the sum of PE and SW:

$$\text{PVA} = \text{PE} + \text{SW} \tag{3}$$

VE is the relationship between PVA and SW and can be described as the proportion of total ventricular mechanical energy that results in external work used to eject blood. Hence, it was calculated as:

$$\text{VE} = \frac{\text{SW}}{\text{PVA}} \tag{4}$$

EEV is an energy efficiency measure calculated as:

$$EEV = \frac{PVA}{SV} \tag{5}$$

MEP produced by the LV per second was calculated as:

$$MEP = SW \times \frac{HR}{60} \tag{6}$$

where HR is the heart rate in beats/min.

$E_{max}$ is a measure of left ventricular systolic function integrating contractility, structural properties, and neurohormonal response [25, 26]. Frequently it is also denoted $E_{es}$ in the literature. The maximal ventricular elastance is the reciprocal of compliance, representing the ability of the myocardium to generate pressure for a given inotropic state and at a given volume. This measure is often referred to as a measure of contractility and was calculated as:

$$E_{max} = \frac{LVESP}{V(T_{LVP,max}) - V_0} \tag{7}$$

where $V_0$ was assumed to be zero mmHg.

Afterload is the sum of all forces that oppose blood ejection during systole: systemic vascular resistance, arterial input impedance, and arterial compliance [27]. $E_A$ was conceptualized as a measure of afterload with the advantage of also being derived from the PV loop and using the definition of LVP at maximum elastance, and thus being comparable to $E_{max}$. $E_A$ derived from PV loops can be calculated as:

$$E_A = \frac{LVESP}{LVEDV - LVESV} = \frac{LVESP}{LVSV} \tag{8}$$

where LVEDV is left ventricular end-diastolic volume and LVSV is left ventricular stroke volume.

Total arterial compliance (TAC) is the compliance in the artery system and can be estimated by the ratio of LVSV to pulse pressure [28]:

$$TAC = \frac{LVSV}{PP} = \frac{LVSV}{SBP - DBP} \tag{9}$$

where PP is pulse pressure.

The systemic vascular resistance (SVR) is the static resistance in the systemic blood circulation and was estimated using the following equation:

$$SVR = \frac{MAP}{CO} \tag{10}$$

where MAP is the mean arterial pressure and CO is the cardiac output. MAP was estimated as:

$$MAP = DBP + \frac{PP}{3} \tag{11}$$

Ventricular-arterial coupling is the interaction between the left ventricle and the arterial system [27] and is calculated as:

$$\frac{E_A}{E_{max}} \tag{12}$$

Given Eqs (6) and (7), and that $V_{(TLVP,max)}$ is approximately LVESV, ventricular-arterial coupling can also be expressed as:

$$\frac{E_A}{E_{max}} \approx \frac{LVESP/LVSV}{LVESP/LVESV} = \frac{LVESV}{LVSV} \tag{13}$$

Fig 1 shows a schematic illustration of a pressure-volume loop and the cardiovascular hemodynamics that can be derived from it.

### Statistical analysis

Statistical analyses were conducted using the software IBM SPSS version 25 (SPS inc., Chicago, Illinois, USA). Gaussian distribution was assessed visually using histograms. Continuous variables are expressed as median and interquartile range [IQR] owing to non-Gaussian distribution. Continuous data were compared using Mann-Whitney U test for independent groups and Wilcoxon signed-rank test for paired comparisons. Intra- and interobserver variability were assessed using Bland-Altman analysis [29] in ten subjects during exercise for LVEDV, LVESV, LVM, and PV loop parameters. The intra- and interobserver variability were quantified using the mean differences between observations and the limits of agreement (bias ± 1.96 SD). Two-tailed P-value of <0.05 was defined as statistically significant [12].

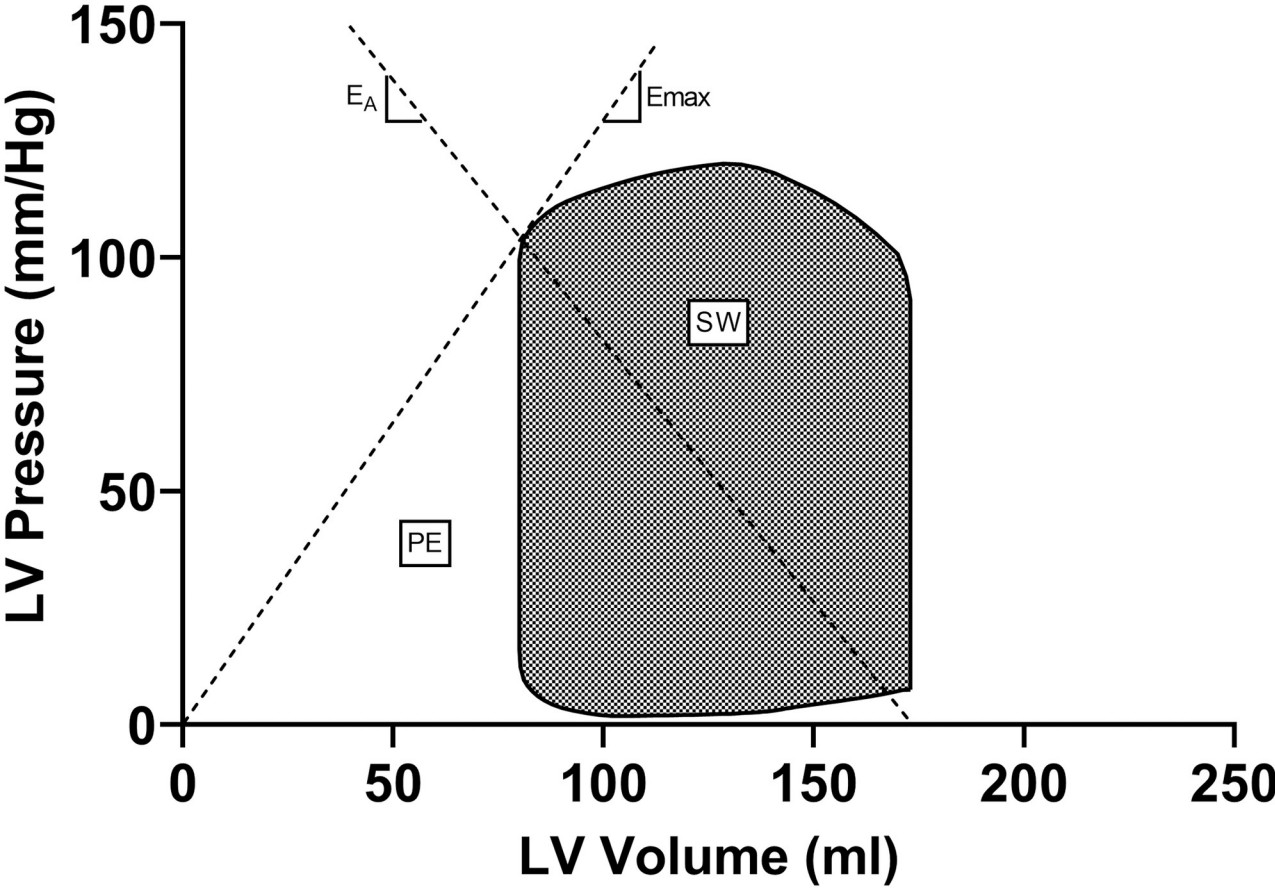

**Fig 1. Schematic figure of a left ventricular pressure-volume loop with cardiovascular hemodynamics marked.** Dashed lines indicate the slopes of maximal ventricular elastance ($E_{max}$) and arterial elastance ($E_A$). $E_A$: arterial elastance; $E_{max}$: maximal ventricular elastance; LV: left ventricular; PE: potential energy; SW: stroke work.

## Results

Baseline characteristics are presented in Table 1. ET had higher training volume (6 h/week) than SC (1 h/week). There were no differences in biometrics (age, length, weight, body mass index (BMI), and body surface area (BSA)) between groups.

### Cardiac volumes at rest and exercise

Cardiac volumes at rest and during exercise are presented in Table 2. Heart rate increased as expected in ET (67% increase) and in SC (41% increase). Systolic blood pressure increased by 10% in ET and by 14% in SC. Also, rate pressure product increased by 92% in ET and by 74% in SC.

ET had higher LVM, LVEDV, and SV compared to SC at rest (Table 2). During exercise, SV were higher in ET than in SC, while LVEDV did not differ between groups. However, LVEDV indexed to BSA differed between groups both at rest and during exercise (Table 2).

The change in cardiac volumes from rest to exercise is summarized in Table 2 and results for LVEDV and LVESV are shown in Fig 2. ET increased SV by 20% from rest to exercise through decreased LVESV and increased LVEDV. SC increased SV by 24% through decreased LVESV, but with no change in LVEDV.

When comparing the response to exercise between ET and SC, there were no differences in delta values from rest to exercise for LVEDV, LVESV, SV, and ejection fraction (Table 2).

### Cardiovascular hemodynamics at rest and exercise

An example of a PV diagram in one athlete at rest and during exercise is illustrated in Fig 3. At rest, ET had lower $E_A$ and higher TAC compared to SC. There were no differences between ET and SC in SW, PE, VE, EEV, $E_{max}$, $E_A/E_{max}$, or SVR (Table 2, Figs 4 and 5).

During exercise, $E_A$ was lower and TAC higher in ET compared to SC, while the remaining hemodynamic measurements (SW, PE, VE, EEV, $E_{max}$, $E_A/E_{max}$, and SVR) did not differ between ET and SC during exercise (Table 2).

For both ET and SC, all hemodynamic variables except PE and EEV differed significantly between rest and exercise. When comparing ET and SC, there were no differences in delta values from rest to exercise in any of the cardiovascular hemodynamics measurements (Table 2).

**Table 1. Subject characteristics.**

|  | ET (n = 13) | SC (n = 10) |
|---|---|---|
| Female (n, %) | 1 (8%) | 1 (10%) |
| Age (years) | 48 [34–60] | 43 [27–57] |
| Height (cm) | 180 [178–186] | 180 [175–184] |
| Weight (kg) | 77 [69–83] | 81 [75–85] |
| BMI (kg/m$^2$) | 23.5 [21.3–25.4] | 24.9 [23.5–25.4] |
| BSA (m$^2$) | 1.96 [1.85–2.08] | 1.99 [1.89–2.09] |
| Training Volume (h/week) | 6 [4.4–7.5] * | 1 [0.4–2] |

Number of females with percentage in parenthesis; other variables expressed as median values [IQR].

BMI: body mass index; BSA: body surface area; ET: endurance-trained subjects; SC: sedentary controls.

* p<0.05 ET vs SC.

**Table 2. Cardiovascular hemodynamics at rest and exercise.**

| | Rest | | Exercise | | Δ | |
|---|---|---|---|---|---|---|
| | ET (n = 13) | SC (n = 10) | ET (n = 13) | SC (n = 10) | ET (n = 13) | SC (n = 10) |
| **Heart rate and pressure** | | | | | | |
| HR (bpm) | 52 [43–59] * | 63 [55–71] | 87 [80–95] † | 89 [83–95] † | 35 (67%*) | 26 (41%) |
| SBP (mmHg) | 113 [104–122] | 114 [109–129] | 124 [118–137] † | 129 [121–140] † | 11 (10%) | 16 (14%) |
| DBP (mmHg) | 70 [61–78] | 65 [62–77] | 74 [66–89] † | 71 [67–84] | 4 (6%) | 6 (10%) |
| RPP (BPM*mmHg) | 5900 [5300–6400] * | 6600 [6200–9300] | 11400 [10100–12300] † | 11500 [10100–12600] † | 5500 (92%*) | 4900 (74%) |
| **Cardiac volumes and function** | | | | | | |
| LVM (g) | 99 [89–118] * | 71 [70–92] | 104 [94–127] *† | 77 [72–98] † | 5 (5%) | 6 (8%) |
| LVMI (g/m$^2$) | 52 [44–58] * | 38 [36–47] | 57 [47–66] *† | 41 [36–47] | 5 (9%) | 3 (8%) |
| LVEDV (ml) | 218 [198–244] * | 191 [180–207] | 233 [211–265] † | 203 [197–233] | 15 (7%) | 12 (6%) |
| LVEDVI (ml/m$^2$) | 111 [105–120] * | 99 [93–104] | 123 [112–127] *† | 107 [99–115] | 12 (11%) | 8 (8%) |
| LVESV (ml) | 98 [84–111] | 92 [82–99] | 94 [74–101] † | 78 [71–89] † | -4 (-4%) | -13 (-15%) |
| LVESVI (ml/m$^2$) | 49 [43–54] | 46 [42–50] | 45 [37–48] † | 40 [36–45] † | -4 (-8%) | -7 (-15%) |
| LVSV (ml) | 124 [103–134] * | 102 [98–107] | 149 [131–164] *† | 130 [119–144] † | 25 (20%) | 28 (24%) |
| LVSVI (ml/m$^2$) | 64 [56–68] * | 52 [49–55] | 77 [71–83] *† | 67 [62–69] † | 13 (20%) | 15 (29%) |
| LVEF (%) | 57 [53–59] | 54 [50–57] | 64 [61–66] † | 63 [60–64] † | 7.6 (13%) | 8.3 (15%) |
| CO (L/min) | 6.3 [5.0–7.5] | 6.6 [5.5–6.9] | 11.0 [9.4–13.7] † | 11.8 [11.1–13.2] † | 4.7 (74%) | 5.3 (80%) |
| CI (L/min/m$^2$) | 3.2 [2.5–3.7] | 3.2 [2.8–3.5] | 5.5 [4.9–6.7] † | 5.9 [5.4–6.6] † | 2.3 (71%) | 2.7 (84%) |
| **Cardiovascular hemodynamics** | | | | | | |
| SW (J) | 1.5 [1.1–1.6] | 1.3 [1.2–1.3] | 1.9 [1.7–2.3] † | 1.8 [1.6–1.9] † | 0.4 (27%) | 0.5 (40%) |
| PE (J) | 0.6 [0.5–0.8] | 0.6 [0.5–0.7] | 0.6 [0.5–0.8] | 0.6 [0.5–0.7] | -0.06 (-9%) | 0 (0%) |
| PVA (J) | 2.1 [1.7–2.4] | 1.9 [1.7–2.0] | 2.4 [2.2–3.1] † | 2.4 [2.1–2.6] † | 0.3 (18%) | 0.5 (25%) |
| VE (%) | 70 [66–73] | 68 [63–72] | 78 [75–80] † | 75 [73–78] † | 7 (11%) | 7 (11%) |
| MEP (J/s) | 1.3 [0.9–1.5] | 1.2 [1.1–1.6] | 2.6 [2.0–2.9] † | 2.7 [2.2–3.0] † | 1.3 (102%) | 1.5 (122%) |
| EEV (J/L) | 16 [15–18] | 18 [16–20] | 17 [15–18] | 18 [17–18] | 0.04 (0.3%) | -0.09 (-0.5%) |
| E$_{max}$ (mmHg/ml) | 1.0 [0.9–1.1] | 1.1 [0.9–1.2] | 1.1 [0.9–1.3] † | 1.2 [1.0–1.3] † | 0.1 (11%) | 0.2 (15%) |
| E$_A$ (mmHg/ml) | 0.8 [0.7–0.9] * | 1.0 [0.9–1.2] | 0.7 [0.7–0.8] *† | 0.9 [0.8–1.0] † | -0.1 (-14%) | -0.1 (-14%) |
| E$_A$/E$_{max}$ | 0.9 [0.8–1.0] | 1.0 [0.9–1.1] | 0.7 [0.6–0.8] † | 0.7 [0.7–0.8] † | -0.2 (-19%) | -0.3 (-30%) |
| SVR (mmHg*min/L) | 14 [11–16] | 13 [11–17] | 9 [7–10] † | 8 [8–9] † | -5 (-36%) | -6 (-43%) |
| TAC (ml/mmHg) | 2.7 [2.4–3.1] * | 2.0 [1.9–2.5] | 82 [77–97] *† | 75 [64–84] † | 79 (30%) | 73 (36%) |

Data expressed as median values [IQR]. Delta values (Δ) are absolute differences between median exercise and median rest, with relative change expressed as percentage in parenthesis.

BPM: beats per minute; BSA: body surface area; CI: cardiac index; CO: cardiac output; DBP: diastolic blood pressure; E$_A$: arterial elastance; E$_{max}$: maximal ventricular elastance; EEV: energy per ejected volume; LVEDV: left ventricular end-diastolic volume; LVEDVI: left ventricular end-diastolic volume index; LVEF: left ventricular ejection fraction; LVESV: left ventricular end-systolic volume; LVESVI: left ventricular end-systolic volume index; ET: endurance-trained subjects; HR: heart rate; LVM: left ventricular mass; LVMI: left ventricular mass index; MEP: mean external power; PE: potential energy; PVA: pressure-volume area; RPP: Rate pressure product; SBP: systolic blood pressure: SC: sedentary controls; LVSV: left ventricular stroke volume; LVSVI: stroke volume index; SVR: systemic vascular resistance; SW: stroke work; TAC: total arterial compliance; VE: ventricular efficiency.

* $p < 0.05$ ET vs SC at corresponding condition (rest or exercise).

† $p < 0.05$ intragroup comparison of rest vs exercise.

## Intra- and interobserver variability

Intra- and interobserver variability of ten subjects during exercise are presented in Table 3. Intraobserver variability was low for key PV loop input variables (LVEDV, and LVESV) and output parameters (SW, E$_A$, and E$_{max}$) (Fig 6). Interobserver variability for LVEDV during exercise was low (4 ± 8%), while for LVESV and LVEF was somewhat larger (21 ± 18% and

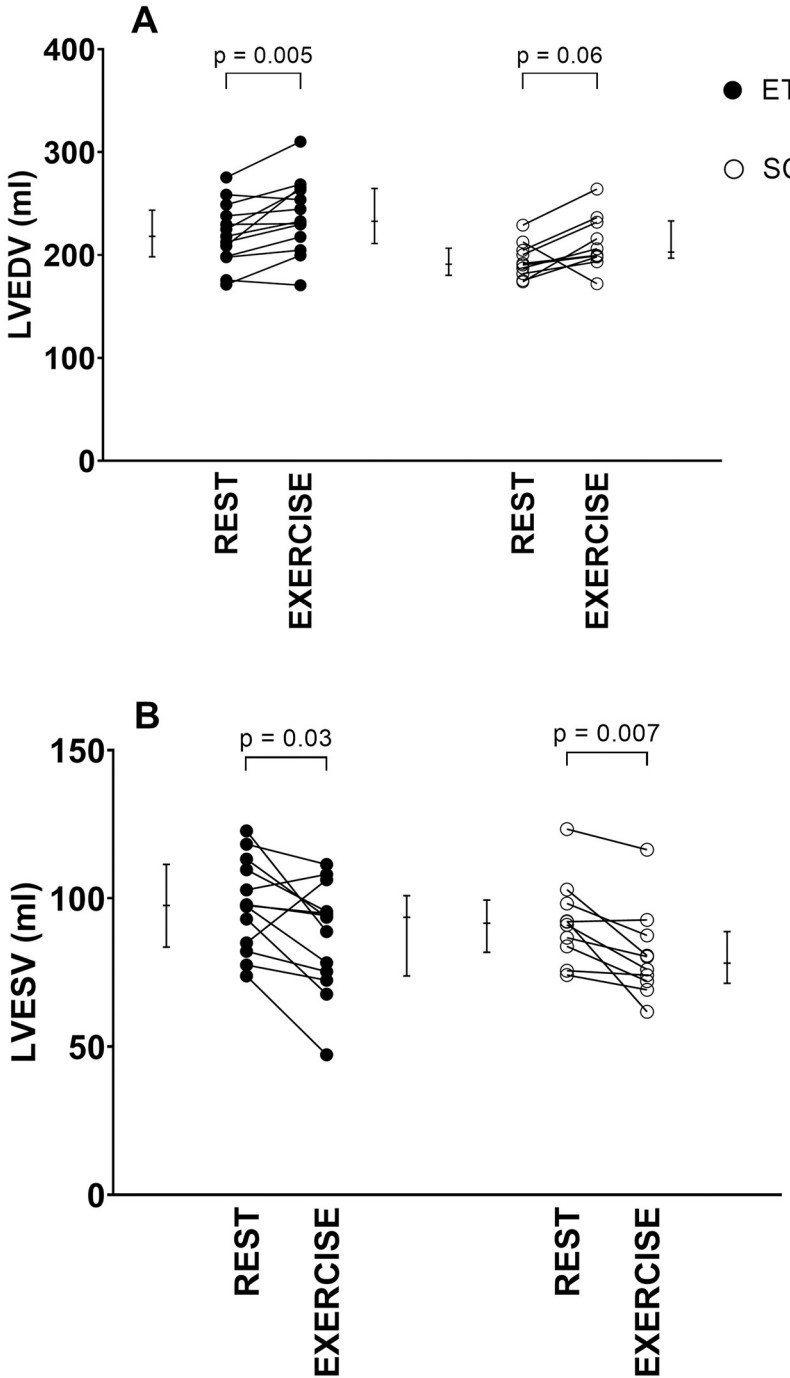

**Fig 2. Left ventricular response in endurance-trained (ET) subjects and sedentary controls (SC) from rest to exercise.** A) left ventricular end-diastolic volume (LVEDV), and B) left ventricular end-systolic volume LVESV). Endurance-trained increased LVEDV and decreased LVESV during moderate exercise compared to rest, whereas sedentary controls had unchanged LVEDV and decreased LVESV. Individual values are plotted, and error bars denote median [IQR]. P-values indicate Wilcoxon signed-rank test comparisons between rest vs exercise in endurance-trained and sedentary controls.

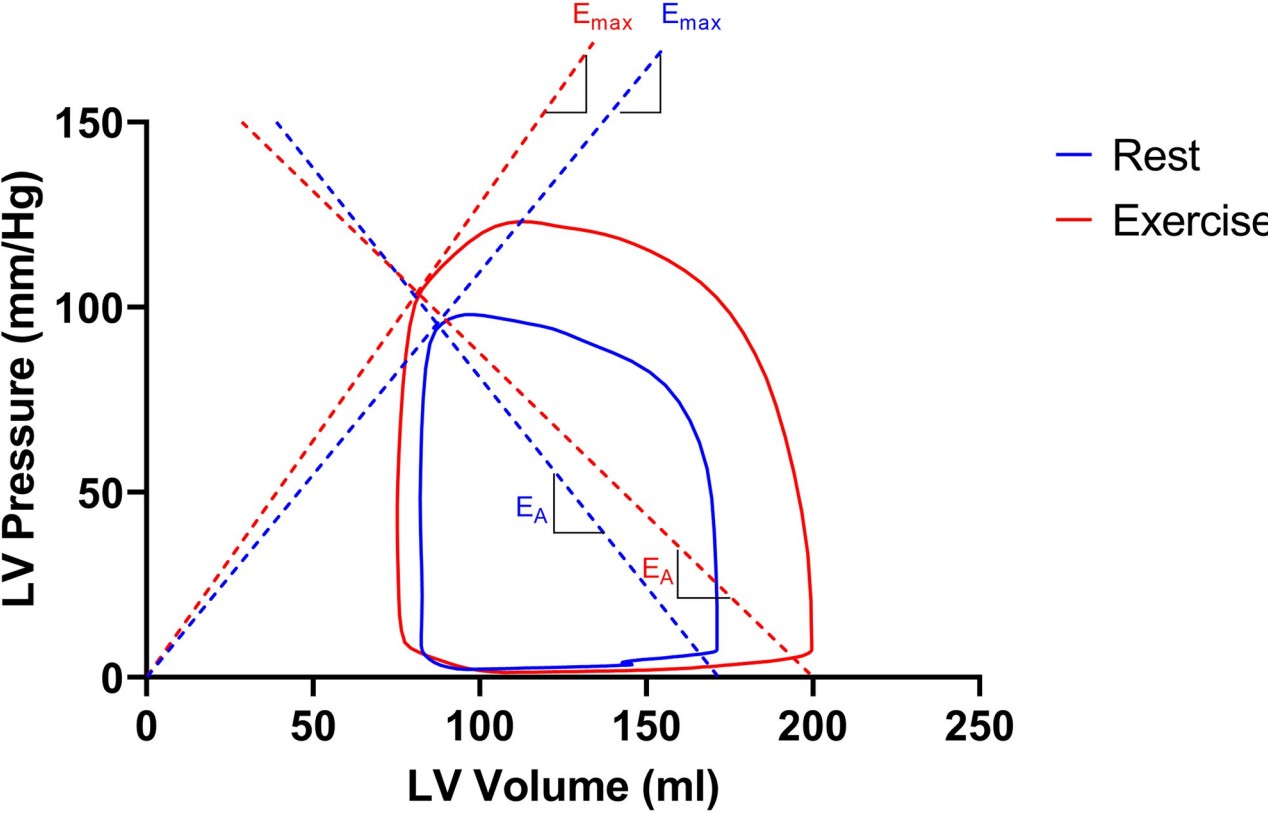

**Fig 3. Left ventricular pressure-volume loop of one endurance-trained at rest (blue) and during exercise (red).** Dashed lines indicate the slopes of $E_{max}$ and $E_A$. Note that $E_{max}$ increases (increased slope) from rest to exercise, whereas $E_A$ decreases (decreased slope), resulting in decrease in ventricular-arterial coupling (left sided displacement of the $E_A/E_{max}$ intercept). Stroke work (i.e. the area within the pressure-volume loop) increases from rest to exercise and ventricular efficiency (i.e. the area within the pressure-volume loop divided by the total pressure-volume area) increases from rest to exercise. $E_A$: arterial elastance; $E_{max}$: maximal ventricular elastance; LV: left ventricular.

-23 ± 18%, respectively). Interobserver variability for PV parameters during exercise are presented in Fig 7.

## Discussion

This is the first study to show feasibility of non-invasive PV loops using CMR during exercise for hemodynamic assessment, adding to prior work by Seemann et al. [9] and Sjöberg et al. [10].

The physiological exercise response of LV volumes are in line with previous studies, demonstrating increased LVSV through increased LVEDV and decreased LVESV in ET, and increased LVSV through maintained or decreased LVEDV and decreased LVESV in healthy volunteers [3, 5, 6]. Furthermore, the physiological exercise response of hemodynamics is also in line with previous studies, demonstrating decreased SVR and increased contractility [30, 31], indicating that non-invasively obtained PV loops during exercise can be used to assess hemodynamic changes. This study contributes to increased understanding of how hemodynamics are affected by moderate exercise, but more importantly presents a novel non-invasive method that can be used to assess cardiovascular hemodynamics during exercise. This method can be of importance for example also in patients with heart failure with preserved ejection fraction (HFpEF) where diagnosis is challenging and may require invasive right heart catheterization [32]. These patients demonstrate LV stiffness [33] and impaired cardiovascular function reserve capacity [34]. A previous study from our group has shown that non-invasive PV

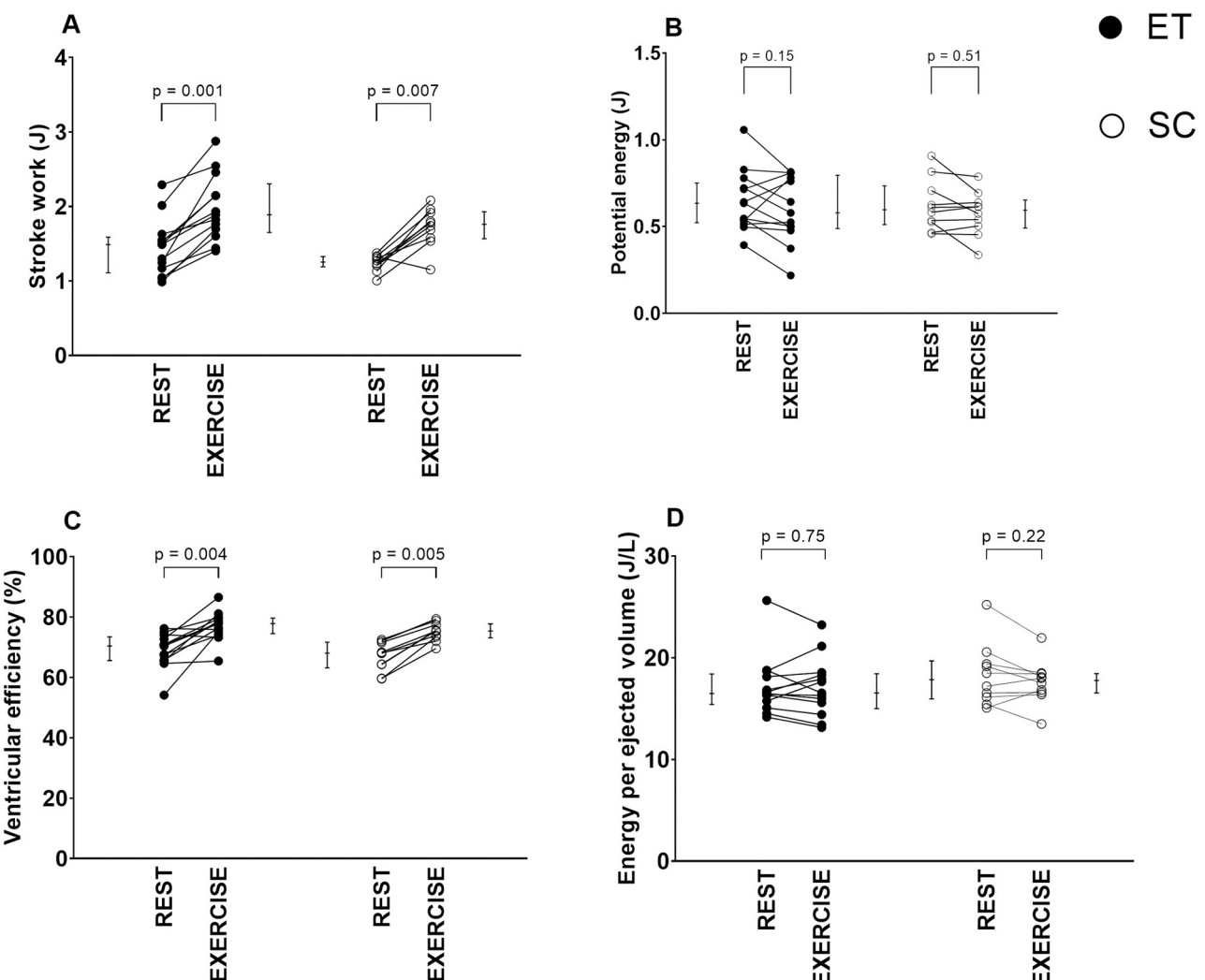

**Fig 4. Change of left ventricular stroke work (SW; panel A), potential energy (PE; panel B), ventricular efficiency (VE; panel C), and energy per ejected volume (EEV; panel D) from rest to exercise.** Endurance-trained (ET) and sedentary controls (SC) did not differ in SW, PE, VE, or EEV at neither rest nor during moderate exercise, and there were no differences in delta values (exercise minus rest) of SW, PE, VE, or EEV between groups. Individual values are plotted and error bars denote median [IQR]. P-values indicate Wilcoxon signed-rank test comparisons between rest vs exercise in ET and SC.

loops at rest are robust across heart failure subtypes [35]. Cardiac MR performed during exercise with non-invasive PV loops may further improve diagnosis and longitudinal evaluations of cardiac function [36].

## Comparing non-invasive and invasive pressure-volume loops

The method of assessing PV loops non-invasively has some advantages compared to the invasive counterpart. Complications of cardiac catheterization include vascular and major incidences, for example hematoma, infection, embolization, and myocardial infarction [37]. In contrast, the method of deriving PV loops non-invasively from exercise CMR comes with the same low risks as an ordinary CMR scan. Although catheterization can provide direct measurements of chamber pressures, the added value of, for example, increased atrial pressures during exercise is limited [38]. While right atrial pressure is demonstrated normal in healthy

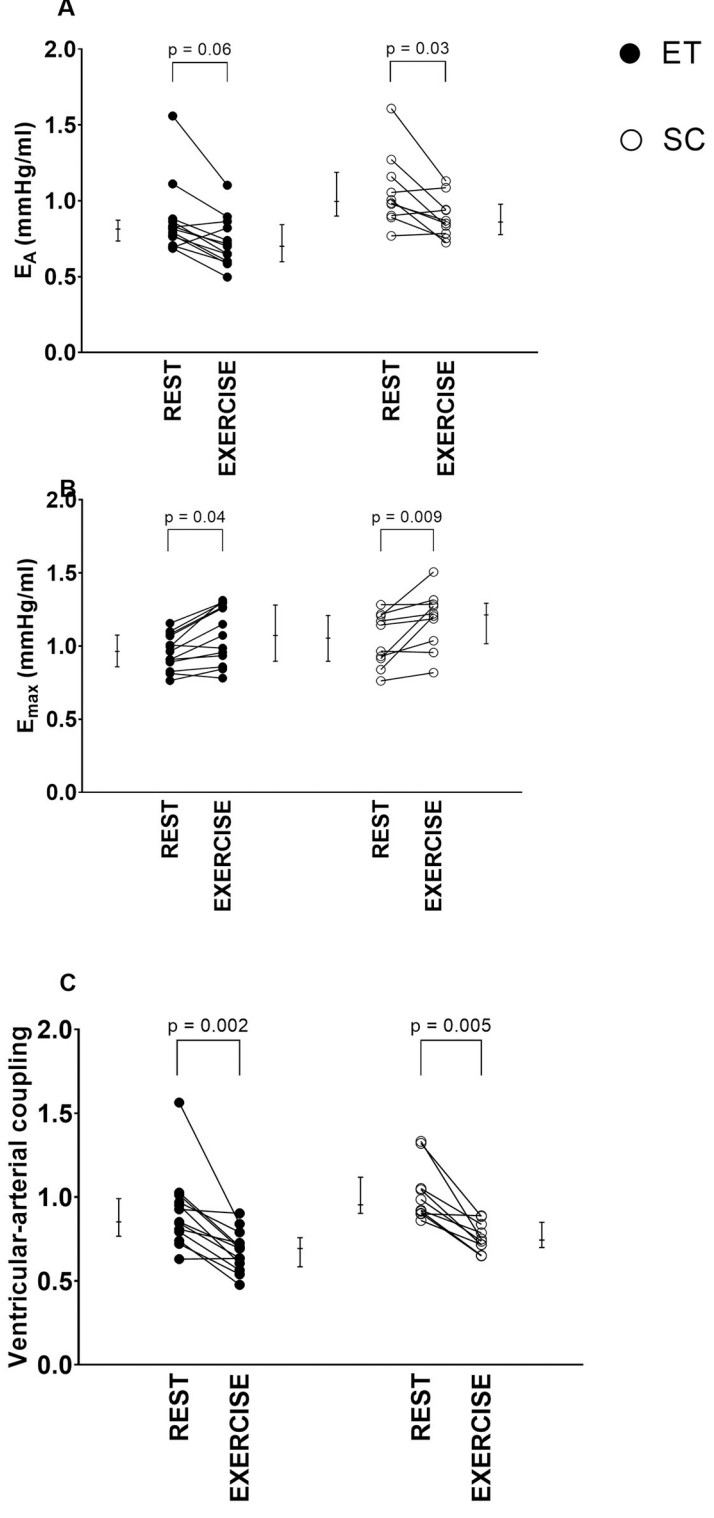

**Fig 5. Change of arterial elastance (E$_A$; panel A), maximal ventricular elastance (E$_{max}$; panel B), and ventricular-arterial coupling (E$_A$/E$_{max}$; panel C) from rest to exercise. Endurance-trained subjects (ET) and sedentary controls (SC) had similar responses in E$_A$, E$_{max}$, and ventricular-arterial coupling to exercise.** Individual values are plotted, and error bars denote median [IQR]. P-values indicate Wilcoxon signed-rank test comparisons between rest vs exercise in ET and SC.

**Table 3. Intra- and interobserver variability of variables during exercise in ten subjects.**

| | Intraobserver variability | Interobserver variability |
|---|---|---|
| LVM (g) | 2 ± 10 g (2 ± 10%) | -26 ± 23 (-23 ± 18%) |
| LVEDV (ml) | 1 ± 14 ml (0 ± 6%) | 10 ± 19 (5 ± 8%) |
| LVESV (ml) | 2 ± 8 ml (2 ± 9%) | 17 ± 15 (21 ± 19%) |
| SW (J) | 0.01 J ± 0.10 (1 ± 5%) | -0.10 ± 0.28 J (-5 ± 16%) |
| PE (J) | 0.01 J ± 0.06 (2 ± 10%) | 0.2 ± 0.1 J (29 ± 27%) |
| VE (%) | -0.2 ± 1.4% (0 ± 2%) | -6 ± 6% (-8 ± 8%) |
| MEP (J/s) | 0.02 ± 0.13 J/s (1 ± 5%) | -0.1 ± 0.4 J (-5 ± 16%) |
| EEV (J/L) | 0.2 ± 0.5 J/L (1 ± 3%) | 0.9 ± 1.2 J/L (5 ± 7%) |
| $E_{max}$ (mmHg/ml) | -0.02 ± 0.05 mmHg/ml (-2 ± 5%) | -0.1 ± 0.2 mmHg/ml (-8 ± 19%) |
| $E_A$ (mmHg/ml) | 0.00 ± 0.04 mmHg/ml (0 ± 4%) | 0.1 ± 0.2 J (13 ± 23%) |

Variability expressed as bias ± 1.96 SD, and as percent in parenthesis.

$E_A$: arterial elastance; $E_{max}$: maximal ventricular elastance; EEV: energy per ejected volume; LVEDV: end-diastolic volume; ESV: left ventricular end-systolic volume; LVM: left ventricular mass; MEP: mean external power; PE: potential energy; SW: stroke work; VE: ventricular efficiency.

individuals during excise, right atrial pressure increases during exercise among patients with pulmonary hypertension, and increased left atrial pressures should be excluded in the diagnosis of precapillary pulmonary hypertension [39]. In addition, the contractile reserve, which can be analyzed from both invasive and non-invasive LV PV loops, provide significant information about cardiac hemodynamics, which is not adequately demonstrated simply by looking at atrial pressures [40]. Thus, non-invasive PV loops from exercise CMR is a safe method providing relevant data about hemodynamics.

## Considerations when using non-invasive pressure-volume loops from exercise CMR

There are some notable strengths and weaknesses with the novel non-invasive PV loop method from exercise imaging proposed in this study. The strengths include first and foremost that PV loops can be calculated non-invasively, providing quantitative data on cardiac hemodynamics during exercise. Secondly, the algorithm computing the PV loops only needs two simple input data: time-resolved volumetrics, and systolic and diastolic blood pressures. Hereby, we provide a feasible non-invasive method to investigate the effect of exercise on cardiovascular physiology. This method may be clinically useful for diagnosis and prognosis of different cardiovascular diseases that can be unmasked by exercise. Exercise CMR is currently used mainly for research purposes, but with technical advancements we expect this method to be increasingly available clinically in the near future.

The weaknesses of the proposed methods are that LVEDP and $V_0$ must be assumed, and that high quality imaging can be challenging at high exercise intensities due to movement of the upper body and heavy breathing. It also needs to be emphasized that the present study was limited to the LV hemodynamics only. There is a need to study right ventricular (RV) physiology in order to increase the understanding of RV pathology. The proposed method could, if tweaked and validated for the RV in a future study, potentially characterize biventricular hemodynamics.

## Exercise effects on cardiac volumes

Cardiac output increased in both groups due to a rise in HR and SV. For both groups, the increase in SV was associated with changes in a combination of load-dependent and load-

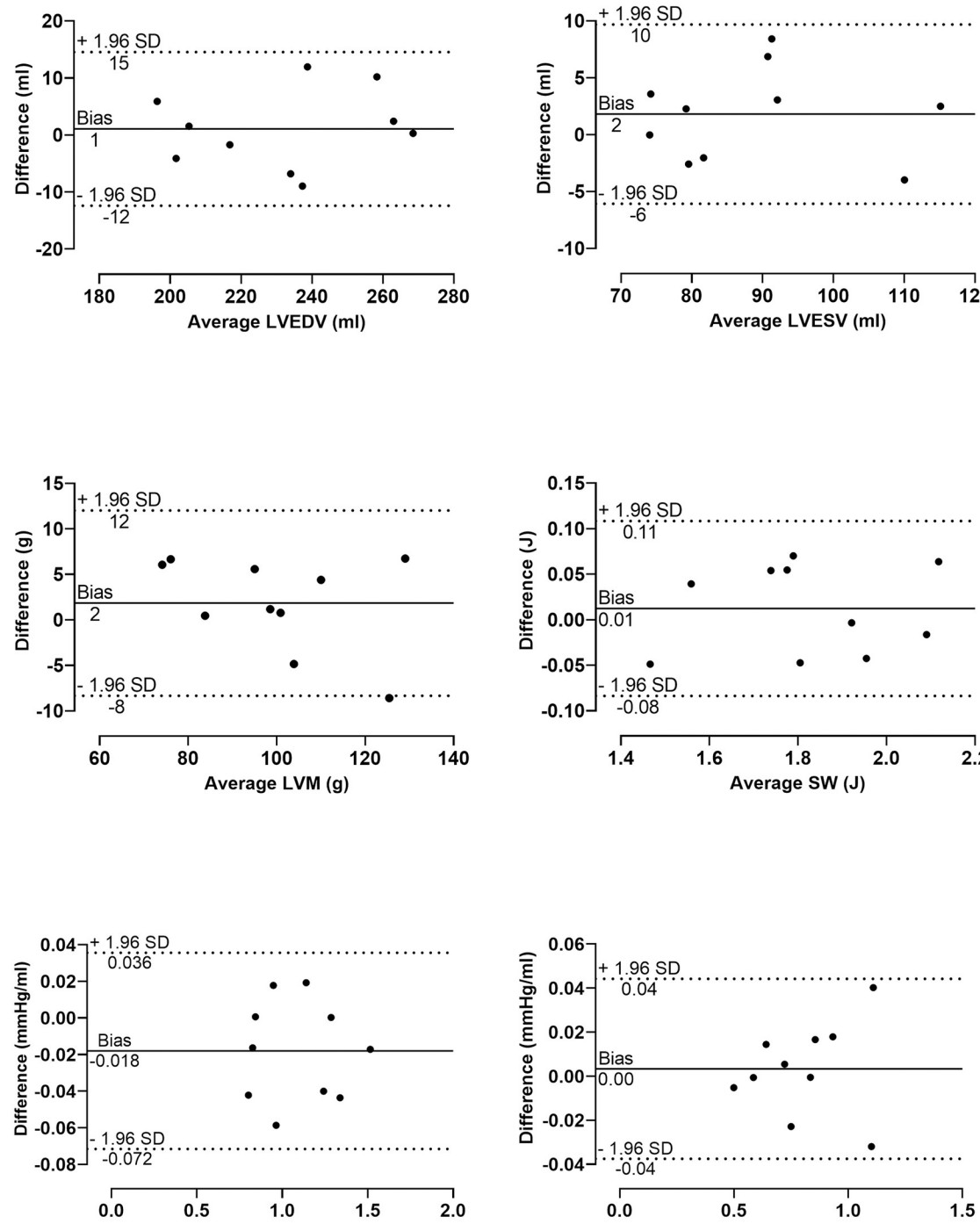

**Fig 6. Bland-Altman plots of intraobserver variability.** Differences between measurements of cardiac magnetic resonance imaging variables and pressure-volume loops parameters during exercise from the same observer across two different time points against the average. Mean of the differences between measurements (bias) and limits of agreement (± 1.96 SD) are marked.

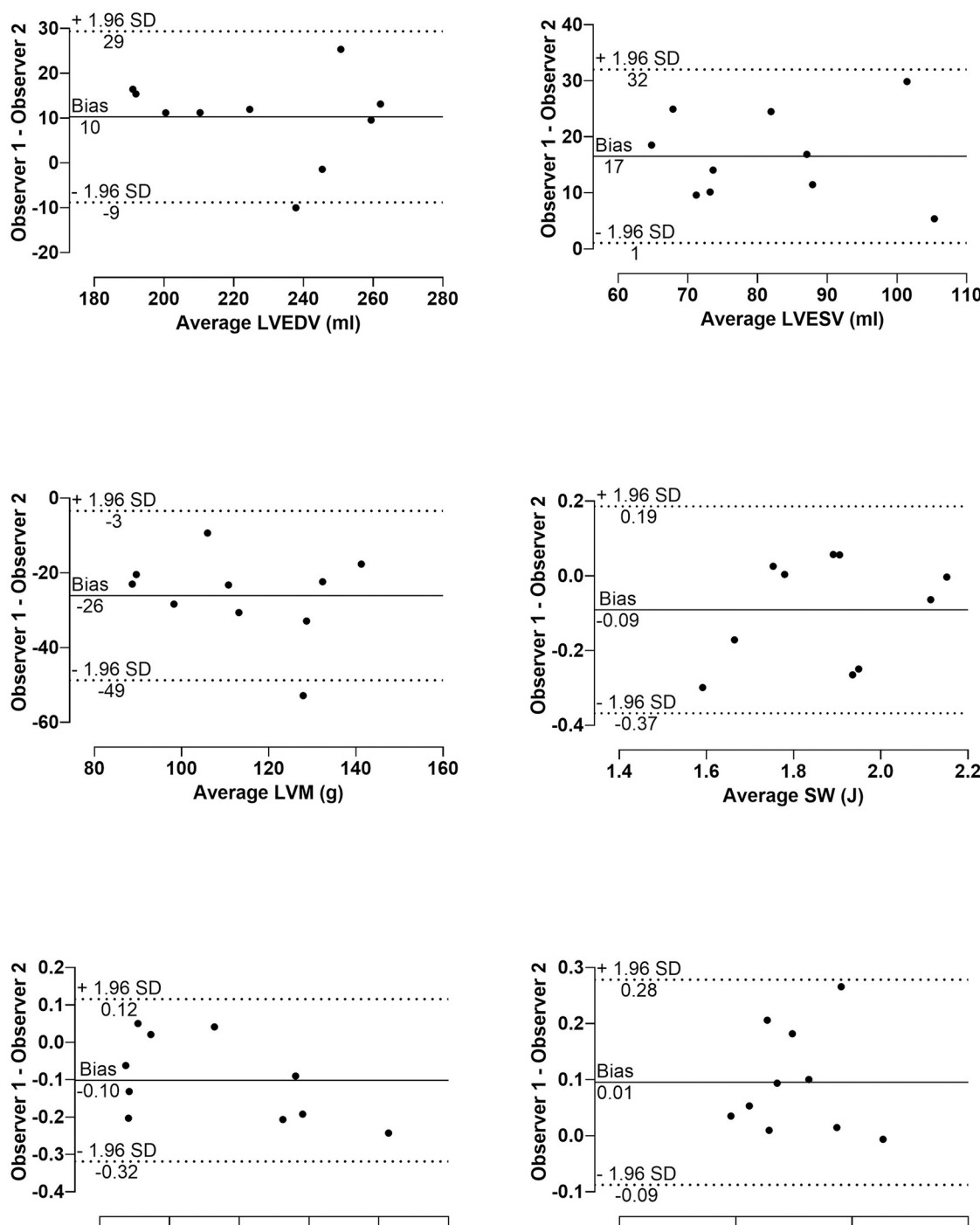

**Fig 7. Bland-Altman plots of interobserver variability.** Differences between observations of cardiac magnetic resonance imaging variables and pressure-volume loops parameters during exercise against the average. Mean of the differences between observers (bias) and limits of agreement (± 1.96 SD) are marked.

independent measures as shown by the load-dependent Starling-effect with a rise in EDV and the load-independent contractility measured as $E_{max}$. When comparing ET and SC, the increase in CO was to a larger extent driven by an increased HR in ET, whereas it for SC was driven by an increased SV. Notably, the increase in EDV in SC was not statistically significant as one subject had, contrary to the other participants, a large decrease in EDV during exercise. This participant's CO increase was driven by increasing heart rate, decreasing LVESV, and increasing contractility measured as $E_{max}$. Of note, this participant was not well-trained (0 hours of endurance training per week) but had the lowest resting HR of all SC (52 bpm), comparable with ET, and the blood pressure reaction was mild (only 105 mmHg systolic blood pressure during exercise).

## Exercise effects on ventricular contractility and arterial elastance

The finding of increased contractility, measured as $E_{max}$, as response to exercise is in line with previously known physiological concepts of sympathetic stimuli increasing ventricular contraction during exercise [41] and with invasive studies investigating the effect of exercise on $E_{max}$ [30, 31, 42, 43]. This supports the feasibility of using CMR images and brachial pressures to calculate PV loops and assess exercise hemodynamics.

The effect on arterial elastance ($E_A$) has differed between previous exercise studies, using both invasive and non-invasive measurements [30, 31, 44]. The results from the present study are in line with Asanoi et al. [30] who found invasive $E_A$ to decrease during aerobic and anaerobic supine cycle exercise in healthy volunteers. On the contrary, $E_A$ has been shown to increase in an intensity-dependent manner in ET athletes [44] and healthy volunteers [31] during cycle exercise, and in subjects with hypertension and normal blood pressure during isometric handgrip exercise [45]. The conflicting results can be explained by the dependency of $E_A$ on the type and intensity of exercise. Hemodynamically, static (isometric) exercise causes pressure load with accompanied increased $E_A$, while dynamic (isotonic) exercise causes volume load with little change in arterial pressure [46]. However, $E_A$ may increase during dynamic exercise at high intensities due to increase in cardiac output [47].

## Ventricular efficiency during exercise

The present study demonstrates increased ventricular efficiency from rest to exercise in both ET and SC. Our data is in keeping with previous studies showing increased ventricular efficiency during exercise invasively in animals [21] and during pharmacological stress non-invasively in humans [10]. However, energy per ejected volume (EEV), which is a metabolic consumption measure, did not change from rest to exercise in ET or SC in the present study. As EEV is the estimated total work of every heartbeat per ml of LVSV, this indicates that ventricular energy efficiency increases with maintained metabolic expenditure per ml of LVSV when ET and SC go from rest to moderate exercise intensity.

## Ventricular-arterial coupling during exercise

The ventricular-arterial coupling ($E_A/E_{max}$) decreased by almost 30% in both groups in response to exercise in our study, consistent with previous invasive and non-invasive studies [30, 31, 48]. Burkhoff and Sagawa constructed a theoretical framework showing that SW is maximized when ventricular-arterial coupling is equal to 1 and peak ventricular efficiency is achieved when ventricular-arterial coupling is equal to 0.5 [49]. The decrease in $E_A/E_{max}$ seen during exercise in our study indicates that, during exercise, the cardiovascular system is approaching optimal ventricular efficiency rather than maximal SW (defined as when $E_{max}$ is equal to $E_A$). Interpreted together with the increase in $E_{max}$, the heart prioritizes ventricular

efficiency at the cost of maximal SW as the ventricular-arterial coupling decreased well below 1 during exercise (Fig 5).

## Hemodynamic changes in athletes versus controls

We did not find any differences between ET and SC in delta values of the hemodynamics in this study. A possible explanation could be that the exercise intensity was only moderate. Increased exercise intensity is associated with decrease in end-systolic volume [4] as well as LV filling and ejection times [50]. ET subjects have higher indexed left ventricular EDV, LVESV, and SV at peak exercise compared to healthy controls [6]. Thus, we expect that the hemodynamics difference in response to exercise would be manifested at higher exercise intensities than achieved in the present study.

## Limitations

Non-invasive PV loops build on assumptions validated at rest [9]. These assumptions might not be fully valid during exercise when the physiological conditions are changed. For example, the assumption that $V_0 = 0$ ml during different hemodynamic conditions has not been validated during exercise, although $V_0$ has been found to be independent of contractile state [25]. It should be noted that this is a theoretical assumption, as the LV volume physiologically can never be 0 ml, and that it is only relevant when considering the diastolic part of the PV loop.

The assumption that LVEDP is constant during exercise has not been validated. However, when developing the non-invasive PV loop-model, Seemann et al. performed a sensitivity analysis of ventricular pressure values ranging from 0–15 mmHg resulting in low bias and variability compared to invasive measures [9]. This sensitivity analysis suggests that if the LVEDP varies within physiological limits during exercise, the impact on PV loop parameters is likely small. It should be noted, however, that there are pathological conditions where LVEDP exceeds 15 mmHg, for example decompensated heart failure [51]. The results from this study may not be generalized to a population with pathological LVEDP.

High heart rates during exercise might induce a confounding factor of the PV loop computations. A refined version of the algorithm taking the relative durations of systole and diastole at high heart rate into account has recently been validated and presented [10]. The refined version of the algorithm was used in the analysis because high heart rates were expected during exercise.

The CMR imaging protocol used included cine balanced steady state free precession sequences, requiring exercise cessation and breath-holds during image acquisition. The consequence of the exercise cessation was a drop in heart rate during image acquisition and although the target heart rate was reached during exercise, image acquisition was actually performed at slightly lower heart rates. Thus, for future studies using exercise cessation the heart rate during exercise should be higher than target heart rate in order ensure image acquisition occurs at the target heart rate. The breath-holds may influence the systolic and diastolic blood pressure, as they have been found increasing with prolonged apnea [52, 53]. However, the breath-holds were typically 8–12 seconds and any apnea-induced increase in blood pressure would therefore be expected to be minor. Exercise was restrained to moderate intensity to ensure good image quality. The current results show that non-invasively obtained PV loops at these intensities can reproduce known physiology. Higher intensities affect image quality by e.g. motion and could subsequently complicate volumetric delineations. This would challenge the results of the PV loop at that intensity. However, as shown by Edlund et al. [54], end-diastolic and end-systolic delineations are reliable at high exercise intensities, and therefore we believe the proposed method to assess cardiac hemodynamics is useful even in that setting.

The study participants exercised in the CMR scanner using a cycle ergometer in the supine position. Supine body position influence hemodynamics by means of changed blood distribution, leading to greater filling pressures compared to upright body position [55]. Supine exercise has less recruited muscle mass and leg perfusion pressure gradient compared to upright exercise, resulting in lower peak $VO_2$, heart rate, and workload [56, 57]. Thus, hemodynamic abnormalities found during upright exercise may not be adequately comparable with supine exercise results using the proposed method from this study.

A limitation to the method, although not a limitation of the study, is the requirement of a CMR scanner with an MRI compatible cycle ergometer. The equipment and adequate expertise are expensive and of limited availability. Additionally, the time required to conduct the test and analyze the data is not insignificant.

## Conclusions

This study shows feasibility of assessing LV PV loops during exercise non-invasively from CMR and brachial pressure. The hemodynamic results are in keeping with invasively demonstrated physiological responses to exercise in healthy subjects. Therefore, our method offers a new useful tool also to assess hemodynamics even during exercise. This opens possibilities to assess hemodynamics in different cardiac disease stages and at higher exercise intensities which has not been possible non-invasively before.

## Supporting information

**S1 Table. Heart rate (BPM) for each image slice.** ET: endurance-trained subject; SC: sedentary control.
(DOCX)

**S2 Table. Systolic and diastolic blood pressures (mmHg) for each exercise bout.** ET: endurance-trained subject; SC: sedentary control.
(DOCX)

## Acknowledgments

The authors would like to thank the study participants for their contribution to the study.

## Author Contributions

**Data curation:** Björn Östenson.

**Formal analysis:** Björn Östenson.

**Funding acquisition:** Einar Heiberg, Katarina Steding-Ehrenborg.

**Investigation:** Björn Östenson, Katarina Steding-Ehrenborg.

**Methodology:** Björn Östenson, Einar Heiberg, Håkan Arheden, Katarina Steding-Ehrenborg.

**Project administration:** Björn Östenson, Katarina Steding-Ehrenborg.

**Resources:** Björn Östenson, Einar Heiberg, Håkan Arheden, Katarina Steding-Ehrenborg.

**Software:** Einar Heiberg.

**Supervision:** Katarina Steding-Ehrenborg.

**Validation:** Björn Östenson, Ellen Ostenfeld, Jonathan Edlund, Katarina Steding-Ehrenborg.

**Visualization:** Björn Östenson, Katarina Steding-Ehrenborg.

**Writing – original draft:** Björn Östenson.

**Writing – review & editing:** Ellen Ostenfeld, Jonathan Edlund, Einar Heiberg, Håkan Arheden, Katarina Steding-Ehrenborg.

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
