## [Decision Letter · Decision Letter 0]

20 Jan 2023

PONE-D-22-29573Endurance-trained subjects and sedentary controls increase ventricular contractility and efficiency during exercise–feasibility of hemodynamics assessed by non-invasive pressure-volume loopsPLOS ONE

Dear Dr. Oestenson,

Thank you for submitting your manuscript to PLOS ONE. After careful consideration, we feel that it has merit but does not fully meet PLOS ONE’s publication criteria as it currently stands. Therefore, we invite you to submit a revised version of the manuscript that addresses the points raised during the review process.

We received three reviews for your manuscript, two with very thorough and proactive comments for improving the manuscript and some questions regarding details and methods of the study. Please use the comments and recommendation by the reviewers to increase the quality of your manuscript and alter your manuscript accordingly. In case you find the requested alterations or concerns not justifiable, please respond and explain in the rebuttal letter. More information about the process is listed below. The third reviewer, who recommended a reject of your manuscript, was concerned about a conflict of interest (one of the authors working for the company who produced the equipment used) and expressed distrust towards the technique. I am sure that this can be addressed by statements and information explaining the influence of the company for the study and also adding some information about the validation of the technique. I therefore recommend to take extra care for dissolving the concerns of this reviewer. 

We look forward to receiving your revised manuscript.

Kind regards,

Hans-Peter Kubis, PD. Dr. rer. nat.

Academic Editor

PLOS ONE

Journal Requirements:

a) Did participants provide their written or verbal informed consent to participate in this study?

b) If consent was verbal, please explain i) why written consent was not obtained, ii) how you documented participant consent, and iii) whether the ethics committees/IRB approved this consent procedure

Reviewers' comments:

Reviewer's Responses to Questions

**Comments to the Author**

1. Is the manuscript technically sound, and do the data support the conclusions?

Reviewer #1: Yes

Reviewer #2: Yes

Reviewer #3: Partly

2. Has the statistical analysis been performed appropriately and rigorously? 

Reviewer #1: Yes

Reviewer #2: Yes

Reviewer #3: Yes

3. Have the authors made all data underlying the findings in their manuscript fully available?

Reviewer #1: Yes

Reviewer #2: Yes

Reviewer #3: Yes

4. Is the manuscript presented in an intelligible fashion and written in standard English?

Reviewer #1: Yes

Reviewer #2: Yes

Reviewer #3: Yes

5. Review Comments to the Author

Reviewer #1: The work conducted by the authors is novel and should be commended. This is a well written manuscript and demonstrates the feasibility of this novel methodology in both trained and untrained participants.

The methods and experimental protocol have been performed by an experienced group and the data presented clearly.

I have no major issues to raise within my review.

My minor points are as follows:

Title:

1 – I would recommend the use of a colon before ‘feasibility of hemodynamics…’. This would be more appropriate here.

Study population:

2 – Please note which sports or modality of endurance training your cohort were undertaking because there can be subtle differences in cardiac adaptations between sports. I.e. swimming vs. land-based exercise, or running vs. rowing.

Cardiac magnetic resonance imaging and exercise protocol:

3 – The use of breath-holds for cardiac image acquisition will briefly increase brachial blood pressures. Indeed, this potentially appears evident in Supplemental table 2 with some fairly large fluctuations in systolic blood pressure between exercise bouts.

It appears that blood pressure measures were taken simultaneously with cardiac image acquisition. Please clarify whether there was a specific order for breath holds and brachial blood pressure measures, whether they were separated, or collected was simultaneously. Additionally, please clarify (if you made note) whether there was consistency of this order between subjects.

4 – What determined one or three slices being acquired, and when the participant began exercise again? I think this should be stated here. Was it a heart rate dropping below 55% of their predicted max, or something different?

Also, were all imaged acquired at a heart rate of 55% predicted maximum or above? Three hearts rates in Supplemental table 1 are below 60 bpm and many are 61-70 bpm. Three incorrect heart rates are also included in the S1 table, two for participant ET 1 and one for ET 7.

Cardiac volumes at rest and exercise:

5 – Inconsistent percentage increases during exercise between the text (line 217) and table 2.

Discussion:

6 – The first part of the sentence starting on line 315 needs to be more specific in order to make the conclusion in the latter part, rather than ‘…in line with widely known and accepted exercise physiology…’. Looking at the references, 3 and 5 appear to contract one another in their findings of SV augmentation via EDV increase or ESV reduction. Given this this section is opening your key findings, I feel that these supporting statements should be clearer as to what you are referring to.

Limitations:

7 – Please add the influence of breath holds on blood pressure as a study limitation as per point 3 of this review.

Thank you for the opportunity to read this manuscript.

Reviewer #2: In this manuscript, Ostenson and colleagues sought to determine whether pressure-volume (PV) analysis of resting and exertional ventricular function can be derived noninvasively from a combination of cardiac MR (CMR) and brachial blood pressure. Thirteen endurance trained athletes and ten controls underwent noninvasive hemodynamic assessment during rest and moderate intensity during supine exercise at 60% maximum predicted heart rate (MPHR). Authors nicely demonstrate the feasibility/success of noninvasively deriving PV loops to facilitate enhanced understanding of ventricular function. Overall, I read this manuscript with enthusiasm. This manuscript has enormous potential; while not yet ready for publication in its current form, I ask the authors to consider comments below as constructive criticisms to further develop what could be a very nice manuscript that would sit very well in the literature, the limitations of this model notwithstanding.

MAJOR COMMENTS

In study population, include how many males/females were included in the ET group, since authors indicate the control population was age/sex matched and included gender data for the controls. These data are included neither in the body of the text nor in table 1.

What do authors mean by “sample size was determined by feasibility”? This is a highly confusing statement. I think the authors mean to indicate that this was an exploratory analysis. The sentence in its current form should be removed and authors should write something to the effect of, “In this pilot analysis, sample size was limited to demonstrate feasibility of non-invasive derivation of PV loops”. Or, remove the sentence entirely.

In abstract, authors indicate that exercise was completed at 60% MPHR, but in the methods section, authors indicate exercise was at 55-70% MPHR. Please clarify what was the exercise intensity for the sake of consistency.

Along these lines, I am confused by the data presented and it’s not clear to me that the intended exercise intensity was achieved. The median age for ET group was 48. 220-48 = 172bpm, so the MPHR for ET group is 172bpm. The HR during exercise (table 2) is 77bpm for this group. 77bpm represents 44% of the MPHR. Authors indicate that exercise was completed at 60% MPHR in the abstract, or 55-70% MPHR in the methods section (see above statement on this discrepancy). A HR of 77bpm is nowhere near 60% MPHR. Please clarify this discrepancy. Similar for the SC group.

Authors nicely and succinctly describe some of the benefits and limitations of noninvasive PV analysis. However, there additional limitations which need to be included. First, is the physiology of supine v. upright hemodynamic assessments. Exercise hemodynamics vary depending on the presence/absence of a gravitational load – ie, exercise in a supine v. upright position, as does work intensity of the skeletal muscle – e.g. reliance on muscle groups during supine exercise that are not engaged to the same intensity during upright exercise. As such, hemodynamic abnormalities occurring during upright exercise likely are not adequately redemonstrated by conducting a hemodynamic assessment in a supine position. This issue really needs to be discussed. The second consideration is cost, since cost associated with cardiac MR is not inexpensive. Along these lines is the time required to complete this type of test and the necessity of having MRI-compatible equipment available – ie, having an ergometer available that is MRI compatible.

As a follow-up, do the authors envision that this technique be used for research purposes only, or for clinical purposes as well? I hope it is the latter, since there is enormous potential for this technique to advance the understanding of mechanisms of functional impairments among individuals with cardiovascular and pulmonary vascular disease – both impacting the LV and RV. For this reason, authors need to expand on the section “Considerations when using non-invasive pressure-volume loops from exercise CMR” to include the above-stated issues.

Authors also need to discuss the feasibility of RV PV analysis, albeit briefly, since the focus of this paper is not on RV hemodynamics, but nonetheless, there have been a large number of manuscripts addressing RV PV analysis, and at least in the United States, both the American Heart Association, and the NIH/NHLBI have published scientific statements in the past several years emphasizing the need to further study and characterize RV physiology/pathophysiology. This manuscript has the potential to sit very well given the current landscape of understanding LV and RV physiology/pathophysiology.

The manuscript would benefit from highlighting the added value of PV analysis compared to traditional invasive hemodynamic analysis with a Swan-Ganz catheter. For example, looking at Tedford’s RV PV exercise data in PH patients (Circ HF 2013, Circ 2016), or Cornwell’s RV PV data in healthy individuals (J. Physiology 2020). Limited data are derived from showing that right atrial pressure increases during exercise among patients with PH and RV dysfunction; similarly, limited insight results from completing an invasive CPET on normal healthy individuals demonstrating that RAP is normal. However, RV PV data in healthy individuals demonstrate significant contractile reserve, which is not adequately demonstrated simply by looking at RAP, or even by a stress echo demonstrating “normal ventricular function”. These examples are meant to be illustrative considerations for the authors to include a discussion on this subject, further supporting their argument as to the potential relevance of the novel method they have described in this manuscript. The discussion section should include mention of these concepts.

The discussion section on afterload (EA) should be elaborated on. Authors should review J. Mitchell’s JACC 1994 paper (JACC 1994; 24: 845-849), demonstrating the differences in hemodynamic response to dynamic v. static exercise. It is not surprising that authors found in this manuscript that Ea was reduced during dynamic exercise.

I disagree with the authors discussion section on VA coupling during exercise. Authors need to be careful about how they define VA coupling; in this manuscript, it is defined as EA/EES. Other authors have reported VA coupling as EES/EA. This is relevant inasmuch as during exercise, EES increases and authors found that EA decreases. This change does not imply, at all, that there is VA uncoupling. Rather, it demonstrates an appropriate augmentation in contractility. However, because in this manuscript, VA coupling is defined as EA/EES, the value of this ratio declines during exercise. however, the decline in this ratio should not lead to the conclusion that the ventricle is becoming uncoupled; if the ratio were defined as EES/EA, then the ratio goes up. Please revise this discussion section appropriately.

Authors should include in the discussion why they chose to study ET for this exploratory analysis. The objective of the study was to demonstrate that PV analysis can be obtained noninvasively. It was not to compare exercise hemodynamics in healthy controls v. elite athletes. Why even include endurance athletes here? from a clinical standpoint, it would have been more informative to generate noninvasive PV loops on patients with unexplained dyspnea, for example, or even heart failure prior to and following a drug intervention. Why were endurance athletes included?

Throughout the manuscript, and in the abstract, specify that you are referring to the left ventricle (LV), since there has been a great deal of emphasis placed on right ventricular (RV) PV analysis – see Tedford’s data from MUSC/Hopkins, Cornwell’s data from Colorado, Brener’s Circ:HF review, etc.

Authors go back and forth between using “ET” for endurance-trained, and spelling out endurance-trained. After the acronym “ET” is introduced the first time, use ET consistently throughout the manuscript.

The limitations section is nicely described – authors nicely emphasize the limitations of their model, particularly the discussion on limitations of assuming Vo = 0. It should also be noted that this assumption is entirely theoretical and in reality, is only relevant when considering the diastolic limb of the pressure-volume curve. It is impossible for pressure to be 0 during systole, so there is no physiologic basis for a Vo = 0 for systole.

Reviewer #3: Review of ”Endurance-trained subjects and sedentary controls increase ventricular contractility and efficiency during exercise–feasibility of hemodynamics assessed by non-invasive pressure-volume loops” by Östenson et al.

In the above-mentioned study, the authors assessed endurance-trained athletes and sedentary controls at rest and during exercise with non-invasive pressure-volume loops generated by CMR and NIBP using Segment (a software from Medviso, Sweden).

The aim of the study was to assess if pressure-volume loops could be obtained during exercise.

Twenty-three subjects, 13 endurance-trained athletes, and 10 sedentary controls matched for age and gender were included. They were examined at rest and at moderate exercise intensity.

Pressure volume loops at rest and during exercise were computed using input data from a combination of volumetric measurements obtained from the CMR data and ESP from non-invasive brachial blood pressure. End-diastolic pressure was inputted by the user.

End-systolic elastance (Ees) and Effective arterial elastance (Ea) were found using single-beat technique.

The authors concluded that the method is feasible and a new useful tool to assess hemodynamics even during exercise.

6. PLOS authors have the option to publish the peer review history of their article (what does this mean?). If published, this will include your full peer review and any attached files.

Reviewer #1: No

Reviewer #2: No

Reviewer #3: No

---

## [Author Response · Author response to Decision Letter 0]

25 Mar 2023

Please see response to reviewers in the attached file titled "Response to Reviewers".

---

## [Decision Letter · Decision Letter 1]

27 Apr 2023

Endurance-trained subjects and sedentary controls increase ventricular contractility and efficiency during exercise: feasibility of hemodynamics assessed by non-invasive pressure-volume loops

PONE-D-22-29573R1

Dear Dr. Oestenson,

We’re pleased to inform you that your manuscript has been judged scientifically suitable for publication and will be formally accepted for publication once it meets all outstanding technical requirements.

Kind regards,

Hans-Peter Kubis, PD. Dr. rer. nat.

Academic Editor

PLOS ONE

Additional Editor Comments (optional):

Reviewers' comments:

Reviewer's Responses to Questions

**Comments to the Author**

1. If the authors have adequately addressed your comments raised in a previous round of review and you feel that this manuscript is now acceptable for publication, you may indicate that here to bypass the “Comments to the Author” section, enter your conflict of interest statement in the “Confidential to Editor” section, and submit your "Accept" recommendation.

Reviewer #1: All comments have been addressed

2. Is the manuscript technically sound, and do the data support the conclusions?

Reviewer #1: Yes

3. Has the statistical analysis been performed appropriately and rigorously? 

Reviewer #1: Yes

4. Have the authors made all data underlying the findings in their manuscript fully available?

Reviewer #1: Yes

5. Is the manuscript presented in an intelligible fashion and written in standard English?

Reviewer #1: Yes

6. Review Comments to the Author

Reviewer #1: Thank you for addressing my review comments so thoroughly and I would like to congratulate you on a well-written, and novel paper.

7. PLOS authors have the option to publish the peer review history of their article (what does this mean?). If published, this will include your full peer review and any attached files.

Reviewer #1: No

---

## [Editor Report · Acceptance letter]

2 May 2023

PONE-D-22-29573R1 

Endurance-trained subjects and sedentary controls increase ventricular contractility and efficiency during exercise: feasibility of hemodynamics assessed by non-invasive pressure-volume loops 

Dear Dr. Östenson:

I'm pleased to inform you that your manuscript has been deemed suitable for publication in PLOS ONE. Congratulations! Your manuscript is now with our production department. 

Kind regards, 

on behalf of

Dr. Hans-Peter Kubis 

Academic Editor

PLOS ONE